# Perspectives in MicroRNA Therapeutics for Cystic Fibrosis

**DOI:** 10.3390/ncrna11010003

**Published:** 2025-01-12

**Authors:** Alessia Finotti, Roberto Gambari

**Affiliations:** 1Department of Life Sciences and Biotechnology, Section of Biochemistry and Molecular Biology, University of Ferrara, 44121 Ferrara, Italy; 2Research Center on Innovative Therapies for Cystic Fibrosis, University of Ferrara, 44121 Ferrara, Italy

**Keywords:** microRNAs, miRNA therapeutics, cystic fibrosis, CFTR, inflammation, bacterial infection

## Abstract

The discovery of the involvement of microRNAs (miRNAs) in cystic fibrosis (CF) has generated increasing interest in the past years, due to their possible employment as a novel class of drugs to be studied in pre-clinical settings of therapeutic protocols for cystic fibrosis. In this narrative review article, consider and comparatively evaluate published laboratory information of possible interest for the development of miRNA-based therapeutic protocols for cystic fibrosis. We consider miRNAs involved in the upregulation of CFTR, miRNAs involved in the inhibition of inflammation and, finally, miRNAs exhibiting antibacterial activity. We suggest that antago-miRNAs and ago-miRNAs (miRNA mimics) can be proposed for possible validation of therapeutic protocols in pre-clinical settings.

## 1. Introduction

Cystic fibrosis (CF) is an hereditary genetic disease caused by a dysregulation of the cystic fibrosis transmembrane regulator (*CFTR*) gene by a chronic hyperinflammatory state and by frequent and severe bacterial infection of the lungs [1,2,3]. The discovery of the involvement of microRNAs (miRNAs) in cystic fibrosis (CF) [4,5,6,7] has generated increasing interest in the past years, due to the possible employment as a novel and very promising class of drugs able to mimic or inhibit miRNA functions. These novel drugs are expected to be employed in pre-clinical settings of therapeutic protocols for cystic fibrosis [8,9]. In this narrative review article, we considered and comparatively evaluated published laboratory information of possible interest for the development of miRNA-based therapeutic protocols for cystic fibrosis. We considered the miRNAs involved in the upregulation of CFTR, the miRNAs involved in the inhibition of inflammation and, finally, the miRNAs exhibiting antibacterial activity. We suggest that molecules exhibiting ago-miRNA and anti-miRNA activity can be proposed for possible validation of therapeutic protocols in pre-clinical settings. The motivation for studying miRNA therapeutics for CFTR upregulation in cystic fibrosis is based on the fact that CFTR correctors and modulators are beneficial for about 80% of the patients with cystic fibrosis [10,11]. Therefore, for an important number of CF patients, therapeutic interventions to modulate CFTR should be implemented. In addition to these considerations, anti-inflammatory therapies and approaches against bacterial resistance to antibiotic therapy are still major challenges in the management and treatment of CF patients [12,13]. In this respect, studies on novel therapeutic approaches, including those based on miRNA targeting or mimicking, are highly demanded.

## 2. MicroRNA Therapeutics: From Laboratory Investigations to Clinical Trials

It is well established that microRNAs are very important non-coding RNAs directing the post-transcriptional regulation of gene expression both in normal and pathological tissues [9,14,15,16]. Several review articles are available describing the biogenesis of miRNAs, their processing, and the mechanisms of translational suppression or degradation of target mRNAs [17,18,19,20]. Micro RNAs are a class of small, single-stranded non-coding RNAs that function as a guide molecule in RNA silencing and hence modulate gene expression [17,18,19,20]. The discovery of the deep involvement of miRNAs in several human pathologies has generated interest in pre-clinical studies, demonstrating the possible application of the so-called “MicroRNA Therapeutics” for the development of clinical protocols [21,22,23]. MicroRNA therapeutics can be divided into at least two major categories: one based on inhibiting miRNA activity, and the second based on mimicking the miRNA biological activity (Figure 1).

### 2.1. The Anti-miRNA Approach: Counteracting miRNAs Causing Pathological Conditions

The significant progress in understanding the molecular basis of human pathologies has revealed that several miRNAs are up-regulated when pathological tissues are compared to their normal counterparts. This is well established in cancer research, where several onco-miRNAs and metastatic-miRNAs have been demonstrated to be deeply associated with cancer [24,25,26,27]. In this particular field of investigation, a large number of review articles and pioneering research studies are available. This approach is based on several preliminary phases: (a) in general, the “pathological miRNAs” are up-regulated in pathological cells or tissues isolated from the patients [28]; (b) “therapeutic” molecules able to interfere with the activity of the “pathological miRNAs” should be available and validated “in vitro” for the highly efficient hybridization to the target miRNA [29]; (c) efficient delivery systems should be available to allow efficient penetration of the anti-miRNA molecules to target cells [30]. This strategy is described in Figure 2.

In the scheme outlined in Figure 2, suitably delivered antagomiRNAs interact within the cell with complementary target miRNAs, causing the inhibition of the molecular interactions between these regulatory miRNAs and the binding sites present within the 3′-UTR of the target mRNA. This causes a decrease in the activity of miRNAs in down-regulating mRNAs, which is associated with increased protein production. This experimental approach has been applied to up-regulate the expression of tumor-suppressor genes, whose expression is down-regulated in many cancer types. One of the most intriguing and interesting examples is the interplay between *PTEN*, miR-221-3p, and antisense molecules against miR-221-3p. *PTEN* is among the most studied and well-characterized tumor suppressor genes [32]. It encodes PTEN (phosphatase and tensin homologue), a tumor-suppressor protein that antagonizes the phosphoinositide 3-kinase (PI3K)-mTOR) pathway through its lipid phosphatase activity, as reviewed by Song et al. [33]. Loss or down-regulation of *PTEN* is a hallmark of several cancers [34,35]. Remarkably, the onco-miR miR-221-3p (hyper-expressed in many cancer types) is able to interact with the 3′-UTR of *PTEN* mRNA, leading to PTEN down-regulation and tumor onset and progression [36,37,38]. Interestingly, anti-miR molecules targeting miR-221-3p have been demonstrated to exert pro-apoptotic and anti-tumor activities both in vitro [39,40] and in vivo [41]. In addition, Xue et al. demonstrated that the treatment of colorectal carcinoma cells with anti-miR221 was able to sensitize the cells to radiation through the activation of *PTEN* [42]. From a drug-development perspective, the anti-miRNA approach should be considered with caution, as a single miRNA is able to interact with tens or even hundreds of potential target mRNAs [43,44]. Another parameter to consider with great attention is the delivery of anti-miRNA molecules in order to achieve the highest biological activity [44,45].

Another important parameter of the antisense approach is efficiency recognition. In this context, an important alternative/complementary strategy is based on the use of “small RNA zipper” molecules able to connect miRNA molecules end to end, as described by Meng et al. [31]. The interaction between these small RNA zippers and miRNAs generates highly stable RNA-RNA duplexes. In these experimental conditions, miRNA activity is blocked. Using this approach to target the oncomiR miR-221, Meng et al. were able to demonstrate that the miR-221 zipper was able to reverse the oncogenic function of miR-221 in breast cancer cells, restoring the activity of mRNAs that were down-regulated in breast cancer cells by miR-221.

### 2.2. The “miRNA-Masking” Approach: Inhibiting the Molecular Interactions Between miRNAs and 3′-UTR miR-Binding Sites

One of the alternative strategies for altering miRNA functions is, rather than hybridizing with the microRNAs, to cover the miRNA-binding site present in the 3′UTR of the target mRNAs, therefore preventing miRNA/mRNA recognition. In these experimental conditions, miRNA function is deeply hampered. This strategy, called miRNA-mask or miRNA-masking, is based on the use of a single-stranded 2′-O-methyl-modified oligoribonucleotide (or other chemically modified molecule), that is fully complementary to the miRNA binding site to be “masked”, usually present and functionally active within the 3′-UTR of the mRNA to be modulated [46]. This technology was described and discussed by Pagoni et al. and by Wang et al. [47,48], and a similar concept was the basis of a study reported by Choi et al. [49]. The most important steps of this approach are summarized in Figure 3.

In the scheme outlined in this figure, suitably delivered miRNA-masking molecules specifically interact within the cell with complementary miRNA-binding sites present within the 3′-UTR of mRNA, causing inhibition or suppression of the molecular interactions of these regulatory miRNAs to the “masked” binding sites. This causes a decrease in the activity of regulatory miRNAs in down-regulating mRNAs, which is associated with an increased protein production.

### 2.3. MicroRNA Therapeutics: The miRNA Mimicking Approach

In several diseases, pathological genes are up-regulated, which is associated with a high content of their respective mRNAs. Examples include mRNAs coding oncoproteins [50] or, in the case of COVID-19, pro-inflammatory genes coding proteins involved in the COVID-19 “Cytokine Storm” [51]. In most of these cases, miRNAs regulating these up-regulated genes are down-regulated. The treatment of target cells with pre-miRNAs mimicking the activity of these down-regulated miRNAs might be of great interest. The objective of the miRNA therapeutics based on the miRNA mimicking approach is to replace the biological activity of these down-regulated miRNAs.

The experimental approach outlined in Figure 4 can be employed in order to restore the biological activity of miRNAs that are down-regulated or fully suppressed in human pathologies. This is, for example, the case of tumor-suppressor miRNAs targeting mRNAs coding oncoproteins [52,53]. As schematically presented in Figure 4, in representative studies, pre-miRNAs are delivered to target cells, where they generate mature miRNAs that are able to interact with the 3′UTR of the mRNAs to be modulated, causing a sharp inhibition of protein production. Synthetic RNA duplexes containing chemical modifications complexed in lipid or polymer-based vectors are used for therapeutic purposes to enhance stability and cellular absorption. Another method involves the use of expression vectors containing the sequence of the miRNA whose levels need to be increased, under the control of a strong promoter. As a representative example, Phatak et al. studied miR-214-3p, an important tumor suppressor in several cancers [54,55,56]. In particular, they studied the interplay between miR-214-3p and the oncoprotein RAB14 in esophageal cancer cells. The major result of this study was that the forced expression of miR-214-3p was associated with a marked decrease in cellular migration and invasion [56].

### 2.4. Pre-Clinical and Clinical Studies Based on miRNA Therapeutics

Several studies (both pre-clinical and clinical) support the concept that miRNA therapeutics deserves great attention for translating laboratory research into clinical practice. In fact, the therapeutic potential of miRNAs for various human pathologies (including cystic fibrosis) is evident, and future studies are expected to validate their possible applicability in clinical settings. The growing interest in these studies is demonstrated by the involvement in miRNA therapeutics of several biopharmaceutical companies, such as Santaris Pharma (Hersholm, Denmark), Roche Pharmaceuticals (Singapore), Regulus Therapeutics (San Diego, CA, USA), Mirna Therapeutics Inc. (Carlsbad, CA, USA), miRagen Therapeutics (Boulder, CO, USA), and EnGeneIC (Sydney, Australia). The involvement of these companies has been reviewed by Zhang et al., 2021 [44]. Examples of biopharmaceutical products for miRNA therapeutics of human diseases are reported in Table 1.

### 2.5. Combining miRNA Therapeutics with Chemotherapy

Several studies have demonstrated that the protocols of miRNA therapeutics can be employed together with chemotherapeutic agents [67,68,69]. For instance, Zurlo et al. combined the treatment of U251 and T98G glioma cells with a peptide nucleic acid targeting miR-221-3p and an anti-tubulin tetrahydrothieno [2,3-c]pyridine derivative and found synergistic activity [67]. A similar approach was followed by Gasparello et al. using sulforaphane and a peptide nucleic acid targeting miR-15b-5p [68]; furthermore, Zurlo et al. found that a combined treatment of glioblastoma U251 cells with an anti-miR-10b-5p exhibits synergistic acidity when combined with an anticancer agent based on 1-(3′,4′,5′-trimethoxyphenyl)-2-aryl-1H-imidazole scaffold [69]. Another example is the demonstration that miRNA therapeutics can overcome cancer resistance to therapy. Studies indicate that combining therapeutic miRNAs with chemotherapy can decrease the required drug doses for cancer treatment [45].

## 3. Pathophysiology of Cystic Fibrosis: Identification of Target Pathological Networks for the Development of Therapeutic Protocols

In cystic fibrosis, the defects of the cystic fibrosis (CF) transmembrane conductance regulator (CFTR) are caused by more than 2000 mutations of the *CFTR* gene, causing deep alterations in CFTR protein content and functions. These quantitative and/or qualitative alterations deeply affect the homeostasis of chloride, bicarbonate, sodium, and water in the airway surface liquid [1,2,70,71,72]. As reviewed by Cabrini et al., this influences the mucus composition and viscosity and is associated in CF patients with severe conditions of infection and inflammation throughout their whole lives [73]. CF is a multi-organ pathology characterized by the following major pathological conditions: (a) a deep dysregulation of the CFTR [74]; (b) a hyperinflammatory state characterized by an elevated expression of pro-inflammatory genes, such as the interleukin-8 (*IL-8*) gene; [75] (c) bacterial infection, the most relevant being that caused by *Pseudomonas aeruginosa* [76]. Micro RNAs are involved in all these pathological conditions, as will be discussed in the following sections.

## 4. MicroRNAs and Expression of the *CFTR* Gene

In this field of investigation, we should mention several research papers that have significantly contributed to the notion that CFTR expression and functions are regulated by microRNA. For example, Gillen et al. reported in 2011 evidence suggesting an involvement of microRNAs in regulating *CFTR* gene expression [77]. In 2012, Ramachandran et al. identified and described a microRNA network that regulates the expression and biosynthesis of wild-type and DeltaF508 mutant *CFTR* [78]. In Table 2, we report a list of miRNAs that interact with the 3′UTR of *CFTR* mRNA and regulate *CFTR*. Summarizing, the miRNAs involved in *CFTR* regulation are, in addition to miR-145 and miR-494 (reported by Gillen et al.) [77], miR-101 [79], miR-144 [79], miR-223 [80], miR-509 [81], miR-384 [82], miR-200b [83], miR-143-5p [84], miR-335-5p [85], miR-16 [86]. It should be mentioned that these microRNAs can synergize, as demonstrated by Megiorni et al. while studying the biological effect of miR-101-3p and miR-294 [87] and by Papi et al. [88], who demonstrated a synergistic enhancement of the *CFTR* gene expression following the combined treatment of bronchial epithelial cells with antisense molecules targeting miR-145-5p and miR-101-3p. The experimental approaches to sustain the role of these miRNAs on *CFTR* gene expression are usually the following: (a) the presence of the miRNA-binding sites within the 3′UTR of the *CFTR* mRNA; (b) the demonstration that these miRNA-binding sites are conserved throughout molecular evolution (and therefore are functionally important) when the nucleotide *CFTR* mRNA sequences of different species are compared; (c) the demonstration that pre-miRNAs down-regulated and antago-miRNAs up-regulated luciferase activity in 3′UTR/CFTR)/luc constructs; no or low modulation of luciferase activity should be found in mutated constructs in which the miRNA-binding sites of the 3′UTR/*CFTR* mRNA are mutated in order to suppress the miRNA/(*CFTR*) mRNA interaction. The final evidence that supports the role of the studied miRNAs in *CFTR* gene regulation is that forcing the expression of these microRNAs is associated with deep inhibition of *CFTR* gene expression, as published by Papi et al. [89]. Conversely, and of great interest in proposing “miRNA Therapeutics” for cystic fibrosis, the treatment of target epithelial cells with inhibitors of miRNAs down-regulating *CFTR* is associated with the hyperactivation of *CFTR* gene expression. In this respect, antago-miRNAs based on peptide nucleic acids (PNAs) have been proposed [90] and employed with impactful results by several research groups [91,92,93,94]. One of the most interesting approaches for CFTR upregulation is, in our opinion, targeting miR-145-5p, as generally recognized by concurrent evidences [93,95,96,97,98]. Interestingly, these treatments can be combined to obtain the highest effect on *CFTR* upregulation, as demonstrated by Papi et al., who reported the synergistic enhancement of the expression of the *CFTR* gene after the combined treatment of bronchial epithelial Calu-3 cells with PNAs targeting miR-145-5p and miR-101-3p [88]. Table 2 summarizes examples of miRNAs regulating *CFTR*.

## 5. MicroRNAs and Inflammation in Cystic Fibrosis

CF transmembrane conductance regulator (CFTR)-deficient airway epithelial cells display signaling abnormalities and aberrant intracellular processes, which lead to transcription of inflammatory mediators. Several transcription factors, especially nuclear factor-κB (NF-κB) are activated [99,100]. As mentioned above, the lung environment of CF patients is characterized by high levels of pro-inflammatory cytokines, such as IL-8, IL-6, and TNF-alpha, and decreased levels of anti-inflammatory mediators, such as IL-10, associated with a marked and persistent neutrophil recruitment into the airways. Thus, the possibility of using cytokine modulators to inhibit the exaggerated inflammation in CF represents a potential rational approach (2). After the initial approval of the dual therapy Orkambi^®^ (Lumacaftor/Ivacaftor; Vertex Pharmaceutical, Boston, Mass, USA), interesting data showed that the functional rescue of F508del-CFTR also significantly reduces the mRNA levels of *CXCL8*, *CXCL1*, and *CXCL2* in response to *P. aeruginosa* exposure, highlighting the potential anti-inflammatory properties of the corrector/potentiator combination [101,102].

Recently reported evidence suggest that microRNAs are deeply involved in controlling all the different steps of inflammation in cystic fibrosis. Some microRNAs promote inflammation, while most of the studied microRNAs are potent inhibitors of the expression of genes responsible for CF inflammation (examples in Table 3) [103,104,105,106,107,108,109,110,111,112].

Among pro-inflammatory microRNAs, one example is miR-155, which promotes inflammation in cystic fibrosis by enhancing the expression of IL-8 [103]. In this context, miRNA therapeutics based on the use of anti-miR-155 molecules are expected to inhibit IL-8 expression. On the contrary, many other miRNAs cause a decrease in the production of pro-inflammatory proteins. For instance, Oglesby reported a decrease in IL-8 production associated with the hyperexpression of miR-17 [104]. Ma et al. reported that miR-302b negatively regulates IL-1β production [105], and Fabbri et al. found that miR-93-5p (down-regulated during *P. aeruginosa* infection) inhibits IL-8 protein synthesis and release [106].

With respect to the mechanism of action of the anti-inflammatory miRNAs, the molecular targets are diverse. Fabbri et al. described a direct interaction between miR-93-5p and *IL-8* mRNA [106], while Kalantari et al. reported that miR-718 represses pro-inflammatory cytokine production by targeting *PTEN* [107]. Interestingly, the NF-kB pathway is a target of the activity of several miRNAs. For instance, Bardin et al. reported that miR-199a-3p reduced IKKβ protein expression, thereby inhibiting NF-κB activation and expression of NF-kB-regulated genes [108]. In addition, Ma et al. [105] and Xu et al. [109] reported the direct targeting of *IRAK4* mRNA by miR-302b and miR-93, respectively. A further example of the involvement of miRNAs in the NF-kB pathways is miR-636, which was demonstrated to exhibit a direct interaction with *IL1R1* and *RANK*; these interactions cause a decrease in the levels of IL1R1 and IKKβ proteins, causing inhibition of pro-inflammatory gene expression [110]. A final example is constituted by the effects of miR-93-5p and miR-145-5p on the expression of Toll-like Receptor-4 (TLR-4). In this respect, *TLR-4* mRNA was found to be a direct target of miR-93-5p and miR-145-5p by Gao et al. [111] and Wu et al. [112], respectively. Figure 5 shows the expected molecular targets and mechanism of action of miR-93-5p.

MicroRNA miR-93-5p directly inhibits *TLR4* [112] and *IRAK4* [109], causing indirect inhibition of NF-kB and of the transcription of the NF-kB regulated *IL-8* gene. In addition, miR-93-5p directly interacts with the 3′UTR of *IL-8* mRNA, thereby causing the inhibition of IL-8 production and release.

From a therapeutic point of view, a decrease in the expression of pro-inflammatory genes should be obtained with transfection of cellular model systems with pre-miR-93-5p generating, after cell penetration, a bioactive miR-93-5p molecule. The possible use of miR-93-5p for anti-inflammatory therapy of Cystic Fibrosis is supported by the very interesting results published by Gao et al. [111]. In their study, Gao et al. employed an in vivo mouse model of LPS-induced acute lung injury (LPS-ALI). When the levels of miR-93-5p expression were analyzed in the lung tissues and bronchoalveolar fluid, they were found to be significantly down-regulated. Interestingly, when agomiR-93 was injected into LPS-ALI mice, agomiR-93-mediated decrease of lung injury was noted, associated with suppression of the LPS-induced inflammatory response.

With respect to testing miRNA therapeutics products in animal models prior to clinical trials, we would like to underline that, at least in theory, ago-miRNA and anti-miRNA products targeting genes relevant to CF (for instance the human *CFTR* and *IL-8* genes, as discussed in Section 4 and Section 5) can be assayed in animal models, obtaining informative results, since the sequences of both regulatory miRNAs and those of miRNA-binding sites are conserved throughout the molecular evolution [93,106,111].

## 6. MicroRNAs and the Development of Antibacterial Strategies for Cystic Fibrosis

In cystic fibrosis, the bacterial infection is one of the most leading causes of morbidity and mortality of CF patients. It is well established that *Pseudomonas aeruginosa* (*P. aeruginosa*) is the most frequently involved in bacterial infection [113]. In addition, the interplay between inflammation and bacterial infection in CF should be considered [114,115,116]. For instance, the expression of miR-302b is induced by TLR2 and the TLR4/NK-κB pathway during *P. aeruginosa* infection, and its overexpression activates cytokine release [114]. Moreover, in a paper published in 2016, Li et al. reported that *P. aeruginosa* infection augments inflammation through the miR-301b repression of c-Myb-mediated immune activation and infiltration [115]. A final example outlining the interplay between *P. aeruginosa* infection and inflammation is provided by another study, by Li at al., who found that overexpressed miR-539 exacerbates *Pseudomonas aeruginosa* pneumonia by promoting inflammatory responses [116]. The role of microRNAs in chronic *Pseudomonas* lung infection in cystic fibrosis has been reviewed by Fesen et al. [117], Ye et al. [118], and Kimura et al. [119]. This issue is of great interest, considering that *P. aeruginosa* has been considered by the World Health Organization (WHO) as one of the priority bacteria, requiring extensive research and urgent development of new antibiotic treatments [120].

One of the examples of miRNAs involved in *P. aeruginosa* infection is miRNA-302b, which was demonstrated by Zhou et al. to augment the host defense against bacteria by regulating inflammatory responses via feedback to TLR/IRAK4 circuits [121]. Another example is constituted by the microRNA-302/367 Cluster, which impacts host antimicrobial defense via the regulation of the mitophagic response to *P. aeruginosa* infection [122]. A product for possible applications as an antibacterial agent is MEG3-4, a miRNA decoy that regulates IL-1β to prevent sepsis during lung infection [123].

An interesting study underlying the interplay between *P. aeruginosa* infection and microRNAs has been reported by Lozano-Iturbe et al., who found that the binding of *P. aeruginosa* to CF bronchial epithelial cells deeply alters the composition of the produced exosomes after comparison with healthy control cells [124]. An application of this and similar observations has been reported by Koeppen et al., who demonstrated that let-7b-5p in vesicles secreted by human airway cells is able to reduce biofilm formation and increase antibiotic sensitivity of *P. aeruginosa* [125]. Taken together, the studies presented in this chapter strongly suggest that the microRNAs involved in *P. aeruginosa* infection should be considered as molecular targets for the development of antibacterial approaches.

## 7. Conclusions and Future Perspectives

In this review paper, we have presented and discussed studies supporting the role of microRNAs in the major pathological conditions of cystic fibrosis, i.e., the regulation of CFTR expression, the process of inflammation and bacterial infection, with particular focus on *Pseudomonas aeruginosa*. These studies support the concept that “microRNA therapeutics (see Figure 1, Figure 2, Figure 3 and Figure 4) might be considered in the future for the possible development of therapeutic protocols for cystic fibrosis. Interestingly, clinical trials based on miRNA therapeutics are ongoing for the treatment of human pathologies (Table 1). It is well established that anti-miRNA molecules down-regulating CFTR (for instance PNAs against miR-101-3p, miR-145-5p and miR-335-5p) are able to restore CFTR activity. In this respect, it should be noted that targeting multiple CFTR-regulating miRNAs might lead to high efficiency in CFTR upregulation. This was reported by Papi et al., who combined treatment with PNAs against miR-101-3p and miR-145-5p to obtain maximal enhancement of CFTR expression [88], and by Megiorni et al., who combined miR-101 and miR-494 synthetic mimics, demonstrating synergism in inhibiting the expression of a reporter construct containing the 3′-UTR of *CFTR* in luciferase assays [87]. Some miRNAs are involved in different processes, such as miR-145-5p, which is involved in CFTR regulation [77,81,93,95,96,97] and in inhibiting the expression of pro-inflammatory genes [112].

Important issues that should be considered in future experimental studies should clarify the impact of miRNA therapeutics in comparison with other strategies for modifying gene expression using DNA/RNA-based biomolecules, such as siRNAs and ASO. This comparison is necessary, considering that (a) inhibiting one single miRNA might activate the expression of a large cohort of genes, all negatively controlled by the same miRNA, and (b) that mimicking the activity of a single miRNA might inhibit the expression of many different mRNAs containing the binding site of the miRNA itself. In this respect, transcriptomic and proteomic studies should be highly warranted to verify this possibility. Examples of limitations and challenges of the miRNA therapeutic approach for CF are presented (together with possible solutions) in Table 4.

Other issues of miRNA therapeutics were extensively presented and discussed in two review papers by Zhang et al. [44] and by Seyhan [45]. Among future perspectives, we would like to discuss the issues below.

### 7.1. Combined Treatments

A very interesting field of investigation for the future is the combination of miRNA therapeutics with current chemotherapy. For instance, De Santi was able to demonstrate that CFTR-specific target site blockers (TSBs) masking the miR-145-5p and miR-223-3p binding sites increased CFTR expression and enhanced the effects of ivacaftor/lumacaftor or ivacaftor/tezacaftor [84]. In our opinion, this very important issue deserves to be investigated in great detail, considering that CFTR correctors and modulators are beneficial for about 80% of the patients with cystic fibrosis, but not for the entire population of CF patients [10,11]. Therefore, for an important number of patients, therapeutic interventions to modulate CFTR should be implemented, and combined treatments exploring synergisms between CFTR corrector/potentiators and “miRNA Therapeutics” products might be considered in pre-clinical approaches.

### 7.2. Innovative Diagnostic Tools for a Personalized miRNA Therapeutics of Cystic Fibrosis 

The analysis of the pattern of extracellular circulating microRNAs (EC-miRNA) has been extensively studied in projects focusing on the liquid biopsy of cancer [128,129]. These studies were followed by a number of investigations demonstrating the usefulness of EC-miRNAs as non-invasive biomarkers in a wide range of diseases [130,131]. Although few reports are available on EC-miRNAs in CF, this issue should be deeply investigated, in our opinion, considering the potential importance of miRNA signatures for programming therapeutic interventions, including those based on miRNA therapeutics. It is widely accepted that circulating EC-miRNAs might be important for monitoring the response of patients to therapy. In this respect, Cook et al. demonstrated for the first time that changes in circulating miRNA levels in CF might be predictive of CF-associated complications (hepatic fibrosis), suggesting that serum-based miRNA analysis might be have prognostic value [132]. More recently, Ideozu et al. employed microarray technology to identify aberrantly expressed plasma ECmiRNAs in CF and were able to demonstrate significant differences in the ECmiRNAs pattern when CF patients were compared with non-CF subjects [133]. The results of this study suggest that ECmiRNAs may be clinically relevant in CF. Further studies in this field of investigation are warranted, as also suggested in the review article by De Palma et al., who highlighted recent findings on the potential utility of measuring circulating miRNAs in CF [7]. In particular, it would be very important to find a positive correlation between the analysis of EC-miRNAs and the miRNAs found in vivo in the bronchial epithelium of CF patients, where the increased expression of miRNAs (for instance miR-145, miR-223, and miR-494) correlates with decreased CFTR expression [80]. Although further studies are required to define the importance of EC-miRNAs as diagnostic, prognostic, and predictive biomarkers, the possible interplay between the expression of EC-miRNAs in CF patients and the response to therapy (including miRNA therapeutics) should be considered of top interest.

### 7.3. Delivery

A further major issue (not covered by the present review) is the delivery of miRNA therapeutics molecules. Several studies are ongoing in this specific issue. PNAs can be efficiently delivered to target cells by the attachment of a cell-penetrating 8-arginine peptide [40]. TSBs have been delivered after encapsulation in poly-lactic-co-glycolic acid (PLGA) nanoparticles [84], PNAs and miRNA mimics can be delivered using argininocalix [4] arenes [134,135]. The research on these very important issues is expected to reinforce the possible application of miRNA therapeutics (also in combination with conventional interventions) for the experimental treatment of cystic fibrosis.

## Figures and Tables

**Figure 1 ncrna-11-00003-f001:**
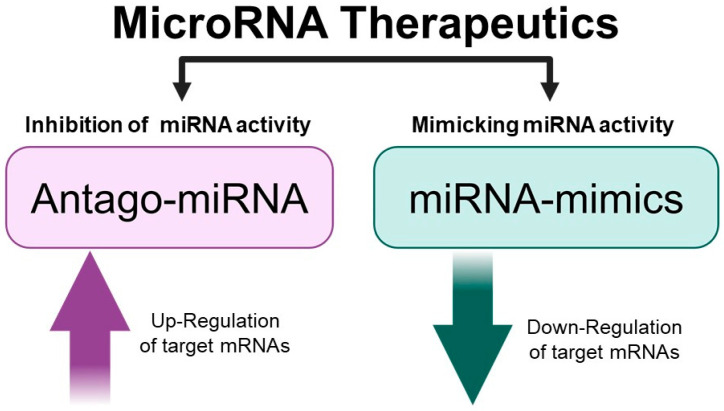
MicroRNA therapeutics. Picture created using Bio-Render.com (7 November 2024).

**Figure 2 ncrna-11-00003-f002:**
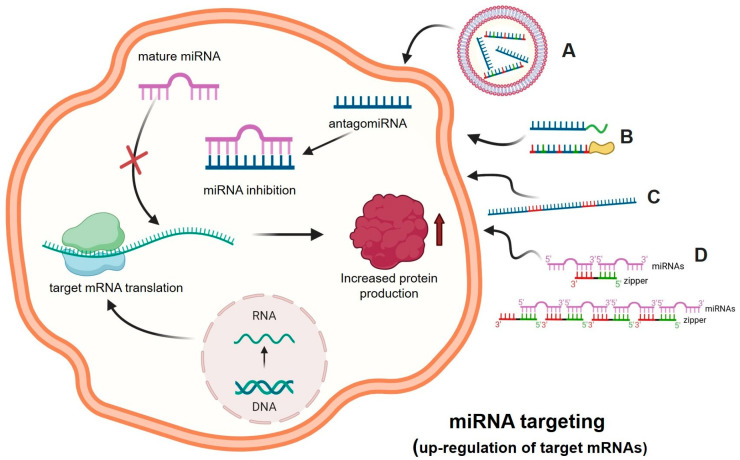
MicroRNA Therapeutics: the anti-miRNA approach. A forced inhibition of the miRNA activity can be obtained using antago-miRNA (anti-miRNA) oligonucleotides (AMOs) (e.g., DNA, RNA, and nucleic acids analogs such as LNA, PNA, 2′-MOE), delivered with vectors (A) or bioconjugated for an increased cellular uptake (e.g., R-PNAs) and/or a targeted delivery (B). MicroRNA inhibition can also be achieved using anti-miRNA sponge RNA sequences that contain multiple microRNA binding sites (C). The approach based on the use of zipper oligonucleotides is shown in panel (D) [31]. The binding between the miRNA and the anti-miRNA molecules leads to the inactivation of the miRNA, as it can no longer bind to its molecular target, i.e., messenger RNA, thus increasing protein production. Picture created using Bio-Render.com (7 November 2024).

**Figure 3 ncrna-11-00003-f003:**
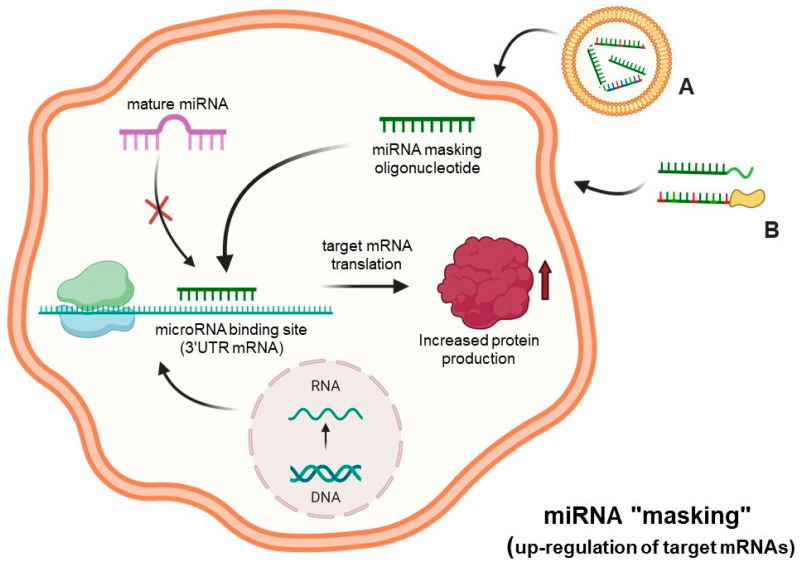
MicroRNA therapeutics: the “miRNA-masking” approach. The down-regulation of microRNA functions is obtained through miRNA masking oligonucleotides and analogs delivered to cells (A,B), which act by masking the miRNAs binding site of target mRNAs through a direct hybridization of the miRNA “mask” with the 3′UTR region of mRNA. Created using Bio-Render.com (7 November 2024).

**Figure 4 ncrna-11-00003-f004:**
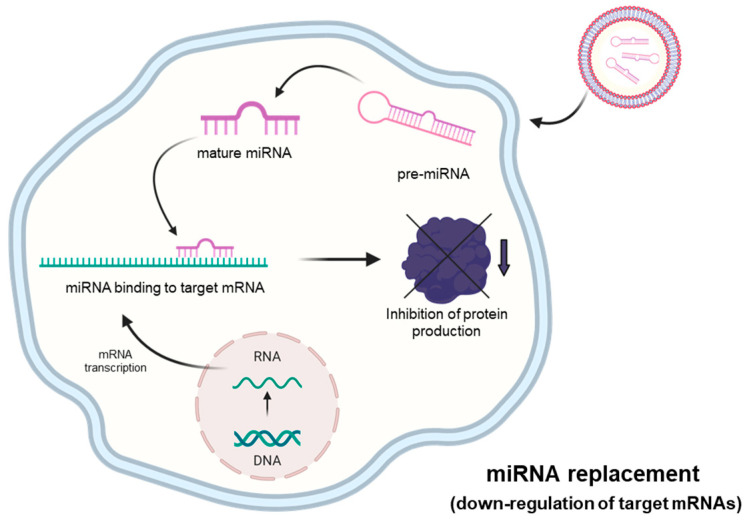
MicroRNA therapeutics: the “miRNA-replacement” approach. This strategy is based on the use of molecules (mature double-strand microRNA mimics or pre-miRNA oligonucleotides) that can restore physiological levels of miRNA with consequent inhibition of mRNA translation. Created using Bio-Render.com (7 November 2024).

**Figure 5 ncrna-11-00003-f005:**
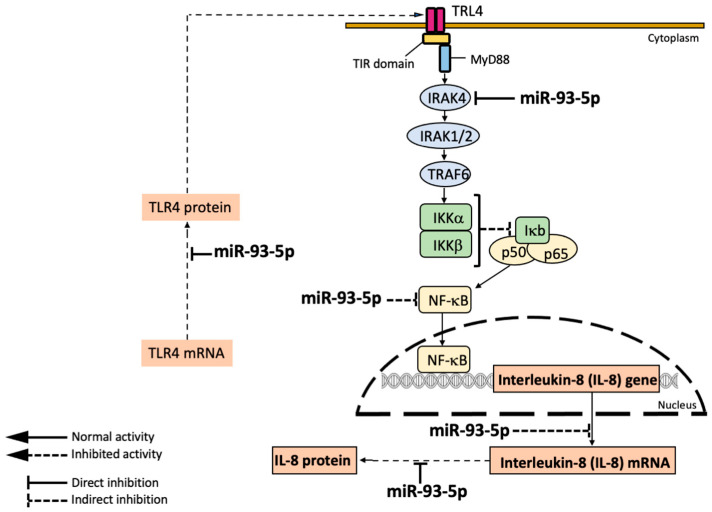
Mechanism of action and molecular targets of miR-93-5p according to the reports published by Fabbri et al. [106], Xu et al. [109], and Gao et al. [111]. MicroRNA miR-93-5p directly interacts with *IL-8* mRNA, thereby inhibiting IL-8 production and release [106]; in addition, miR-93-5p inhibits *IRAK1*, thereby preventing NF-kB activation and expression of NF-kB-dependent genes, such as *IL-8* [109]; in addition, miR-93-5p is able to interact with *TLR-4* mRNA [112], thereby down-regulating the NF-kB pathway.

**Table 1 ncrna-11-00003-t001:** Examples of biopharmaceutical products for miRNA therapeutics targeting human diseases.

Biopharm Product	Molecular Target or Mechanism of Action	miRNA Therapeutic Approach	Target Pathology or Clinical Condition	Pharmaceutical Companies	Clinical Trials	References
Miravirsen (SPC3649)	A 15-mer LNA ASO (PS-modified) targeting miR-122	Antisense	Hepatitis C virus (HCV) infections	Santaris Pharma Roche Pharm (Basel, Switzeerland)	NCT01200420, NCT01872936, NCT02031133, NCT02508090	Huang et al., 2021 [57]
RG-012	An anti-miR-21	Antisense	Alport syndrome	Regulus Therapeutics Inc. (San Diego, CA, USA)	EudraCT: 2016-002181-32	Chavez et al., 2022 [58]
RG-101	An anti-miR-122	Antisense	Hepatitis C virus (HCV) infections	Regulus Therapeutics Inc.	EudraCT: 2013-002978-49	van der Ree et al., 2017 [59]
MRG-201	A synthetic RNA oligonucleotide mimicking miR-29	miRNA mimicking	Hypertrophic scars; idiopathic pulmonary fibrosis.	MiRagen Therapeutics, Inc. (Boulder, CO, USA)	NCT02603224 (completed)	Chioccioli et al., 2022 [60]
MRX34	A synthetic miRNA designed to mimic the activity of the tumor suppressor miR-34a	miRNA mimicking	Melanoma, NSCLC, hepatocellular carcinoma, renal carcinoma	miRNA Therapeutics Inc. (Austin, TX, USA)	NCT01829971 (terminated)	Hong et al., 2020 [61]
Cobomarsen (MRG-106)	An LNA-based antagomir targeting miR-155	Antisense	Lymphoma subtypes; diffuse large B-cell lymphoma	Miragen Therapeutics (Viridian Therapeutics Inc., Waltham, MA, USA)	NCT03837457 NCT02580552, NCT03713320	Cheng et al., 2022 [62]
MRG-110	A synthetic antagomir of miRNA-92a	Antisense	Ischemic conditions (heart failure)	MiRagen Therapeutics	NCT03603431 (completed)	Abplanalp et al., 2020 [63]
Remlarsen (MRG-201)	An LNA RNA mimic of miR-29	miRNA mimicking	Keloid disorder (scar); fibrotic diseases	MiRagen Therapeutics	NCT02603224 (completed) NCT03601052 (completed)	Gallant-Behm et al., 2020 [64]
MesomiR 1	A mimic of miR-16	miRNA mimicking	pleural mesothelioma	EnGeneIC (New York, NY, USA)	NCT02369198 (completed)	Reid et al., 2016 [65]
RGLS4326	A 9-mer ASO fully complementary to the seed sequence of miR-17	Antisense	Autosomal dominant polycystic kidney disease (ADPKD)	Regulus Therapeutics Inc.	NCT04536688	Lee et al., 2019 [66]

**Table 2 ncrna-11-00003-t002:** Examples of miRNAs involved in regulation of CFTR, and frequently up-regulated in CF.

microRNA	Reference	Experimental Strategies Employed to Confirm CFTR Regulation	Comments
miR-145-5p	Gillen et al. [77] De Santi et al. [84] Fabbri et al. [93] Finotti et al. [95] Lutful et al. [96] Sultan et al. [97]	Inhibits expression of a reporter construct containing the *CFTR* 3′ UTR; inhibits the expression of endogenous CFTR; antisense molecules enhance CFTR expression; miRNA masking molecules enhance CFTR expression.	Numerous cell lines were employed, among which 16HBE14o-, Calu-3, IB3-1, Cufi-1, Nuli-1, CaCo-2. In addition, primary human airway epithelial cells were used. The effect of miR-145-5p on CFTR has been confirmed by several studies of different research groups, and pre-clinical studies are highly recommended.
miR-494	Gillen et al. [77]	Directly targeting discrete sites in the *CFTR* 3′ UTR (untranslated region) was demonstrated.	Expressed in primary human airway epithelial cells, where CFTR expression is low.
miR-101	Hassan et al. [79]	When premature miR-101 was transfected in human airway epithelial cells, it directly targeted the *CFTR* 3′UTR and suppressed the expression of the CFTR protein.	16HBE14o-cells have been employed (in vitro assays). In vivo data are available, demonstrating that mice exposed to cigarette smoke for 4 weeks up-regulated miR-101 and suppressed CFTR protein in the lungs.
miR-144	Hassan et al. [79]	Transfection of pre-miR-144 suppressed the expression of the CFTR protein.	In vivo data are available, as described for miR-101.
miR-223	Oglesby et al. [80]	Overexpression and inhibition studies were performed with pre-miRs or anti-miRs, respectively, and a luciferase reporter gene was used to elucidate direct miRNA–target interactions.	Bronchial brushings and bronchial cell lines were studied. Increased expression found in vivo in bronchial epithelium of ΔF508 CFTR patients, correlating with decreased CFTR expression.
miR-509	Ramachandran et al. [78,81]	Human non-CF airway epithelia, transfected with a miR-509-3p mimic, showed decreased CFTR expression.	Co-operates with miR-494 in the regulation of CFTR abundance and function. Primary air–liquid interface cultures of human airway epithelia and the Calu-3 cell line have been employed.
miR-384	Viart et al. [82]	These authors used miRNome profiling and gene reporter assays.	Primary human nasal epithelial cells from healthy individuals and CF patients were employed. The regulation of the switch from strong fetal to very low CFTR expression after birth was studied.
miR-200b	Bartoszewska et al. [83]	Analysis of epithelial cell lines during prolonged hypoxia revealed that CFTR expression decreased, while miR-200b was continuously up-regulated.	Two human airway epithelial cell lines, Calu-3 and 16HBE14o-, were employed. These authors utilized in silico predictive protocols to establish potential miRNAs regulating *CFTR* and identified miR-200b as a candidate molecule.
miR-143-5p	De Santi et al. [84]	Luciferase assays were performed to elucidate direct miRNA–target interactions.	Employed human bronchial epithelial cell lines: 16HBE14o−, (CFBE41o-−; IB3-1, Cufi-1, Nuli-1. This study sustains the development of novel therapeutic strategies for increasing the efficacy of the currently available CFTR modulators.
miR-335-5p	Tamanini et al. [85]	Antisense PNAs enhanced the expression of *CFTR* and *NHERF1* genes, analyzed by RT-qPCR and Western blotting.	Lung epithelial Calu-3 cells were employed. Possible involvement of the CFTR scaffolding protein NHERF1 has been proposed.
miR-16	Kumar et al. [86]	Transfection of miR-16 rescues F508del-CFTR function.	Cell lines and native cystic fibrosis epithelial cells were employed.

**Table 3 ncrna-11-00003-t003:** Examples of miRNAs involved in the regulation of pro-inflammatory gene expression and frequently down-regulated in CF.

microRNA	Reference	Target mRNA	Comments
miR-155	Bhattacharyya et al. [103]	Specifically reduced levels of SHIP1, promoting PI3K/Akt activation	Micro RNA miR-155 is one of few examples in which upregulation of miRNAs is associated with upregulation of pro-inflammatory genes (*IL-8*).
miR-17	Oglesby et al. [104]	*IL-8*	The authors suggest that “Modulating miR-17 expression in cystic fibrosis bronchial epithelial cells may be a novel anti-inflammatory strategy for cystic fibrosis and other chronic inflammatory airway diseases”.
miR-302b	Ma et al. [105]	*IRAK4*	Proposed alteration of the NF-kB pathway
miR-93-5p	Fabbri et al. [106] Xu et al. [109] Gao et al. [111]	*IL-8*, *IRAK4*, *TLR4*	The effect of miR-93-5p on *CFTR* has been confirmed by several studies of different research groups; the key paper by Gao et al. demonstrates the in vivo anti-inflammatory activity of miR-93-5p, and pre-clinical studies are highly recommended.
miR-718	Kalantari et al. [107]	*IRAK1*, *PTEN*	Down-regulation of phosphatase and tensin homolog (PTEN) promotes phosphorylation of Akt, leading to a decrease in pro-inflammatory cytokine production.
199a-3p	Bardin et al. [108]	*IKKβ*	Proposed alteration of the NF-kB pathway
miR-636	Bardin et al. [110]	*IL1R1*, *RANK*	Proposed alteration of the NF-kB pathway
miR-145-5p	Wu et al. [112]	*TLR4*	miR-145-5p is also involved in the regulation of *CFTR* expression (Table 1).

**Table 4 ncrna-11-00003-t004:** Challenges of the “MicroRNA Therapeutics” approach.

Major Challenges/Obstacles	Possible Solutions	Reference(s)
The target miRNA down-regulates multiple genes, and the therapeutic antisense miRNA molecule exhibits unwanted upregulation of a large set of genes, in addition to *CFTR*.	(a) instead of “antisense miRNA Therapeutics” use the miRNA-masking approach to increase specificity [90,91]; (b) consider using the ASO-mediated modulation of translation for CFTR upregulation	Sultan et al. [97] De Santi et al. [98] Sasaki et al. [126]
The employed pre-miRNA molecules down-regulate multiple genes, in addition to IL-8 and other pro-inflammatory genes.	(a) consider the possibility of using siRNAs for mRNA targeting; (b) consider the possibility of using ASO siRNAs for mRNA targeting	Mewa et al. [127]
A single antago-miR is not effective due to the fact that the 3′UTR of the mRNA target contains multiple miRNA binding sites.	(a) consider the possibility of using combined treatments based on different antagomiRNA molecules; (c) consider using the approach based on small RNA zippers to lock miRNA molecules and block multiple miRNAs activity	Papi et al. [88] Meng et al. [31]

## Data Availability

All the data are contained within the article; additional information will be shared upon request to the corresponding authors.

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
