# Peer review of "Perspectives in MicroRNA Therapeutics for Cystic Fibrosis"

_ncrna, 2025, doi:10.3390/ncrna11010003_

Round 1

Reviewer 1 Report

Comments and Suggestions for Authors

The authors Finotti and Gambari have described the role of microRNAs in Cystic Fibrosis (CF) and how these can be developed as therapeutics.

The review article is comprehensive and includes supportive schematic figures.

However, the authors should include the 2 key references:

Ramachandran S., Karp P.H., Jiang P., Ostedgaard L.S., Walz A.E., Fisher J.T., Keshavjee S., Lennox K.A., Jacobi A.M., Rose S.D., et al. A microRNA network regulates expression and biosynthesis of wild-type and ΔF508 mutant cystic fibrosis transmembrane conductance regulator. PNAS. 2012;109:13362–13367. 

Kumar P, Bhattacharyya S, Peters KW, Glover ML, Sen A, Cox RT, Kundu S, Caohuy H, Frizzell RA, Pollard HB, Biswas R. miR-16 rescues F508del-CFTR function in native cystic fibrosis epithelial cells. Gene Ther2015 Nov;22(11):908-16doi: 10.1038/gt.2015.56. Epub 2015 Jul 2. 

  •  

Author Response

REPLY TO REVIEWER #1

First of all, we would like to thank the reviewer for her(his) work.

General comments. The authors Finotti and Gambari have described the role of microRNAs in Cystic Fibrosis (CF) and how these can be developed as therapeutic. The review article is comprehensive and includes supportive schematic figures.

Answer. We thank for the positive comments and for the useful suggestions.

Point 1. However, the authors should include the 2 key references:

Ramachandran S., Karp P.H., Jiang P., Ostedgaard L.S., Walz A.E., Fisher J.T., Keshavjee S., Lennox K.A., Jacobi A.M., Rose S.D., et al. A microRNA network regulates expression and biosynthesis of wild-type and ΔF508 mutant cystic fibrosis transmembrane conductance regulator. PNAS. 2012;109:13362–13367. 

Kumar P, Bhattacharyya S, Peters KW, Glover ML, Sen A, Cox RT, Kundu S, Caohuy H, Frizzell RA, Pollard HB, Biswas R. miR-16 rescues F508del-CFTR function in native cystic fibrosis epithelial cells. Gene Ther. 2015 Nov;22(11):908-16. doi: 10.1038/gt.2015.56. Epub 2015 Jul 2.

Answer to point 1. The study by Ramachandran S.et al. was already cited in the submitted paper (ref.71). However, the work by this research group has been further commented in Table 2. Two papers by Ramachandran are present in the revised version of the manuscript (Ref. n.78 and Ref. n.81). The very important work by Kumar et al. has been also included in Table 2 (Examples of miRNAs involved in regulation of CFTR..). The study has been of course included in the reference list (Ref n.86).

In conclusion, we thank the Reviewer for the work done and we hope that the manuscript will be now considered acceptable for publication on Non-coding RNA.

We are of course willing for further implementation, if necessary.  Waiting for your comments, we thank in advance for your consideration and help.

Professor Alessia Finotti

Professor Roberto Gambari

Department of Life Sciences and Biotechnology

Ferrara University

Ferrara, Italy

Reviewer 2 Report

Comments and Suggestions for Authors

In this review, the authors describe different strategies to inhibit or supplement miRNA activity for treating cystic fibrosis. They begin by describing strategies to inhibit or supplement miRNA activity. They then describe the role of miRNA in regulating CFTR and altering CF pathology. They end by discussing additional technical advances needed to apply this technology for the treatment of cystic fibrosis.  

Overall, the article is written clearly. However, there are a few limitations and addressing these limitations may improve the review.  

  1. One limitation of miRNAs mentioned in the review is that miRNA may downregulate multiple genes. It is unclear why miRNAs should be particularly investigated over other RNA-based agents. For example, siRNAs can be generated to knockdown specific genes and may not suffer from this limitation. Similarly, ASOs targeting the 5’UTR of CFTR have been proposed to upregulate CFTR expression (PMID: 31351782). A discussion on why one might choose miRNAs over these other agents might particularly benefit readers  

  1. In many instances, the review seems to merely restate conclusions from the cited studies. Did these studies differ in the type of cells used (primary vs immortalized cells) or in the use of animal models? If so, is the role of some miRNAs more widely accepted or more credible than others?  

  1. The introduction is not clear on the motivation for investigating miRNA therapeutics for CF over other modalities. For example, what is the need for miRNA therapies to treat cystic fibrosis when there are modulator therapies? Is it to treat modulator non-responsive individuals? Providing readers a stronger motivation might improve interest in the article  

  1. What is the translational pathway for an miRNA therapy? Are there any in clinic for other disorders? Since human miRNA sequences may differ from model organisms, how do we test the efficacy of miRNA-based therapies prior to clinical trials? Are the sequences conserved in mice or other animals? 

Line 305: Issue is of great interest

Author Response

REPLY TO REVIEWER #2

First of all, we would like to thank the reviewer for her(his) work.

General comments. In this review, the authors describe different strategies to inhibit or supplement miRNA activity for treating cystic fibrosis. They begin by describing strategies to inhibit or supplement miRNA activity. They then describe the role of miRNA in regulating CFTR and altering CF pathology. They end by discussing additional technical advances needed to apply this technology for the treatment of cystic fibrosis.  

Overall, the article is written clearly. However, there are a few limitations and addressing these limitations may improve the review.

Answer. We thank for the positive comments and for the useful suggestions.

 Point 1. One limitation of miRNAs mentioned in the review is that miRNA may downregulate multiple genes. It is unclear why miRNAs should be particularly investigated over other RNA-based agents. For example, siRNAs can be generated to knock down specific genes and may not suffer from this limitation. Similarly, ASOs targeting the 5’UTR of CFTR have been proposed to upregulate CFTR expression (PMID: 31351782). A discussion on why one might choose miRNAs over these other agents might particularly benefit readers.

Answer to Point 1. We agree with the reviewer that the miRNA Therapeutic approach should be compared with the several approaches that have been proposed, such as the results of oligonucleotides (and analogues) based therapeutic approaches (as those based on siRNAs and ASO). We addressed this very important issue by including the new Table 4, by slightly amending Figure 2 (including some information on the “zipper” approach, lines 103-111) and by including the sentences “Important issues that should be considered in future experimental activity should clarify the impact of miRNA Therapeutics in comparison with other strategies to modify gene expression using DNA/RNA-based biomolecules …Examples of limitations and criticalities of the miRNA Therapeutic approach for CF are presented (together with possible solutions) in Table 4”. (lines 376-385). We also included the citation to two review papers discussing the limits of miRNA Therapeutics and included the following sentence “Other issues of miRNA Therapeutics have been extensively presented and discussed in two review papers by Zhang et al. [44] and by Seyhan [45]. Among future perspectives, we would like to discuss the following….” (lines 388-390).   

Point 2.  In many instances, the review seems to merely restate conclusions from the cited studies. Did these studies differ in the type of cells used (primary vs immortalized cells) or in the use of animal models? If so, is the role of some miRNAs more widely accepted or more credible than others?

Answer to Point 2. We thank the reviewer for her(his) comment. We agree on the fact that convergent studies from different laboratories on the effects of miRNA inhibitors or miRNA mimics should be underlined in our review, especially in the cases where the miRNA therapeutic intervention has been done in vivo. Information on the use of cell lines, primary cells and/or in vivo model systems have been presented in Table 2 (in the column “Comments”). Furthermore, we have included the following sentence to comment on the very interesting results by Gao et al on miR-93: “The possible use of miR-93-5p for anti-inflammatory therapy of Cystic Fibrosis is

supported by the very interesting results published by Gao et al. [105]. In their study, Gao et al. employed an in vivo mouse model of LPS-induced acute lung injury (LPS-ALI). …. when agomiR-93 was injected to LPS-ALI mice, agomiR-93-mediated decrease of lung injury was noted, associated with suppression of the LPS-induced inflammatory response” (lines 310-317). The same article was briefly commented in Table 3: “…. the key paper by Gao et al. demonstrates the in vivo anti-inflammatory activity of miR-93-5p”. We believe that miR-145-5p (as regulatory of CFTR) and miR-93-5p are of great interest, being studied by different laboratories with convergent results.

Point 3.  The introduction is not clear on the motivation for investigating miRNA therapeutics for CF over other modalities. For example, what is the need for miRNA therapies to treat cystic fibrosis when there are modulator therapies? Is it to treat modulator non-responsive individuals? Providing readers a stronger motivation might improve interest in the article.

Answer to Point 3. Thank you for this suggestion. We have included in the Introduction a short paragraph on CF (page 1, lines 21-23) and a sentence clarifying the motivation to propose miRNA Therapeutics for CF: “The motivation for studying miRNA-therapeutics for CFTR upregulation in Cystic Fibrosis is based on the fact that CFTR correctors and modulators are beneficial for about 80% of the patients … In this respect, studies on novel therapeutic approaches (including those based on miRNA targeting or mimicking) are highly demanded” (lines 34-42).      

Point 4.  What is the translational pathway for an miRNA therapy? Are there any in clinic for other disorders? Since human miRNA sequences may differ from model organisms, how do we test the efficacy of miRNA-based therapies prior to clinical trials? Are the sequences conserved in mice or other animals?

Answer to point 4. A partial list of Clinical Trials based on miRNA Therapeutics products has been presented in column #6 of Table 1. Interestingly, pre-clinical studies on a variety of animal systems are possible and highly informative, since miRNA sequences and the respective miRNA binding sites are conserved throughout molecular evolution. This was discussed by including the following sentence: “With respect to testing miRNA Therapeutics products in animal models prior to clinical trials, we would like to underline that, at least in theory ago-miRNA and anti-miRNA products targeting genes relevant for CF (for instance the human CFTR and IL-8 genes, as discussed in chapters 4 and 5) can be assayed in animal models, obtaining informative results, since the sequences of both regulatory miRNAs and miRNA binding sites are conserved throughout the molecular evolution [93,107,112]” (lines 318-323).   

In conclusion, we thank the Reviewer for the work done and we hope that the manuscript will be now considered acceptable for publication on Non-coding RNA..

We are of course willing for further implementation, if necessary.  Waiting for your comments, we thank in advance for your consideration and help.

Professor Alessia Finotti

Professor Roberto Gambari

Department of Life Sciences and Biotechnology

Ferrara University

Ferrara, Italy

Reviewer 3 Report

Comments and Suggestions for Authors

This is a nicely-written comprehensive review article on the current knowledge on miRNAs in cystic fibrosis (CF). The Authors clearly summarized the general (non-CF) and CF-related aspects of abnormal expression of miRNAs as a part of the pathomechanism of the disease, and the therapeutic modification of miRNAs as novel potential treatment solutions. The manuscript contains well-designed figures and tables that aid the understanding of the large amount of data accumulated in the last couple of years in this field.

I have some minor suggestions to improve the manuscript:

1) I am wondering if the trend/change of abnormal level of mentioned miRNAs in CF could be visualized in Table 2 and Table 3 to make it clearer what miRNA alterations can be found in this disease.

2) Only one paper was cited in terms of the association between altered miRNAs and CFTR-modulator treatment. I am wondering if there were more articles in the literature to highlight this key aspect of miRNAs as well.

3) I think the laboratory investigations of circulating miRNAs as potential biomarkers to monitor CF could also be added to this paper to make it more comprehensive.

4) In the subchapter of 2.2, on page 3 and line 113, I think that would be "Figure 3" in stead of "Figure 2". Please, correct it.

Author Response

REPLY TO REVIEWER #3

First of all, we would like to thank the reviewer for her(his) work.

General comments. This is a nicely-written comprehensive review article on the current knowledge on miRNAs in cystic fibrosis (CF). The Authors clearly summarized the general (non-CF) and CF-related aspects of abnormal expression of miRNAs as a part of the pathomechanism of the disease, and the therapeutic modification of miRNAs as novel potential treatment solutions. The manuscript contains well-designed figures and tables that aid the understanding of the large amount of data accumulated in the last couple of years in this field I have some minor suggestions to improve the manuscript.

Answer. We thank for the positive opinion on our study.

Point 1. I am wondering if the trend/change of abnormal level of mentioned miRNAs in CF could be visualized in Table 2 and Table 3 to make it clearer what miRNA alterations can be found in this disease.

Answer to point 1. The microRNAs down-regulating the CFTR (see Table 2) are usually up-regulated in CF; on the contrary, the microRNAs down-regulating the inflammation-related mRNAs (see Table 3) are usually down-regulated in CF. We briefly mentioned this concept in the caption of Tables 2 and 3.

Point 2. Only one paper was cited in terms of the association between altered miRNAs and CFTR-modulator treatment. I am wondering if there were more articles in the literature to highlight this key aspect of miRNAs as well.

Answer to point 2. The cited paper (De Santi et al.) was the best example in studies focusing on combined treatments exploring synergisms between CFTR correctors/potentiators and “miRNA Therapeutics products”. To comment this, we have added the following sentence; “In our opinion this very important issue deserves to be investigated in great detail, considering that CFTR correctors and modulators are beneficial for about 80% of the …. synergisms between CFTR corrector/potentiators and “miRNA Therapeutics products might be considered in pre-clinical approaches” (lines 396-402).

Point 3. I think the laboratory investigations of circulating miRNAs as potential biomarkers to monitor CF could also be added to this paper to make it more comprehensive.

Answer to Point 3. In order to follow the reviewer’s suggestion, we have included the sub-chapter 7.2. Innovative diagnostic tools for a personalized miRNA Therapeutics of Cystic Fibrosis. The analysis of the pattern of extracellular circulating microRNAs (EC-miRNA) has been extensively studied ……. (lines 404-429). Thanks for this suggestion

Point 4. In the subchapter of 2.2, on page 3 and line 113, I think that would be "Figure 3" in stead of "Figure 2". Please, correct it.

Answer to Point 4. You are right. We have corrected this error. Thanks.

In conclusion, we thank the Reviewer for the work done and we hope that the manuscript will be now considered acceptable for publication on Non-coding RNA..

We are of course willing for further implementation, if necessary.  Waiting for your comments, we thank in advance for your consideration and help.

Professor Alessia Finotti

Professor Roberto Gambari

Department of Life Sciences and Biotechnology

Ferrara University

Ferrara, Italy
